# Safety and Efficacy of Autologous Stem Cell Treatment for Facetogenic Chronic Back Pain

**DOI:** 10.3390/jpm13030436

**Published:** 2023-02-28

**Authors:** Ralf Rothoerl, Junee Tomelden, Eckhard Udo Alt

**Affiliations:** 1Isar Klinikum, Munich Sonnenstr 24–26 Department of Neuro/Spine Surgery, 80331 München, Germany; 2Heart and Vascular Institute, Department of Medicine, Tulane University Health Science Center, Tulane University, 1430 Tulane Ave., New Orleans, LA 70112, USA

**Keywords:** facet joint syndrome, adipose tissue-derived regenerative cells

## Abstract

Background: Chronic back pain due to facet joint syndrome is a common and debilitating condition. Advances in regenerative medicine have shown that autologous unmodified adipose tissue-derived regenerative cells (ADRC) provide several beneficial effects. These regenerative cells can differentiate into various tissues and exhibit a strong anti-inflammatory potential. ADRCs can be obtained from a small amount of fatty tissue derived from the patient’s abdominal fat. Methods: We report long-term results of 37 patients (age 31–78 years, mean 62.5) suffering from “Facet Joint Syndrome” The pathology was confirmed by clinical, radiological examinations and fluoroscopically guided test injections. Then, liposuction was performed. An amount of 50–100 cc of fat was harvested. To recover stem cells from adipose tissue, we use the CE-certified Transpose RT™ system from InGeneron GmbH. The cells were then injected under fluoroscopic control in the periarticular fat. Follow-up examinations were performed at 1 week, 1 year, and 5 years. Results: Every patient reported improved VAS pain at any follow-up (1 week, 1 year, and 5 years) with ADRCs compared to the baseline. Conclusions: Our observational data indicate that facet joint syndrome patients treated with unmodified adipose tissue-derived regenerative cells experience improved the quality of life in the long term.

## 1. Introduction

Low back pain (LBP) poses an economic burden to society, mainly in terms of the large number of workdays lost by a small percentage of patients who develop chronic forms of low back pain. A whole variety of pathological processes are responsible for low back pain. One of the leading anatomical structures involved in the pathogenesis of low back pain are the facet joints. Lumbar spinal facet joints were first suggested in the medical literature as a source of low back and lower extremity pain in 1941 [1]. Since then, the so-called “facetogenic back pain” has become a widely accepted diagnosis, though still controversial in the literature [2,3,4,5,6,7,8]. The most substantial support comes from investigations reporting successful back pain relief following peri-articular joint injections [9,10]. Estimates of lumbar facet joint pain prevalence based on single diagnostic blocks have ranged from 7.7% to 75% among patients reporting back pain [11]. The facet joint syndrome is a degenerative process of the facet joint’s cartilage following degeneration of an intervertebral disc that leads to height loss of the segment, resulting in a mechanically induced overload and arthrosis of the respective facet joints (Figure 1). These small joints are made for flexibility but not for weight bearing. Mechanically induced chronic weight overload finally results in a chronic inflammatory process involving the immune system, producing local inflammatory reactions with the synthesis of several pro-inflammatory cytokines and metalloproteinases [12]. Due to the inflammatory nature of the disease, local injection of glucocorticoids into the affected joint has become a standard treatment option. However, several studies suggest that such injections represent no long-term treatment for patients with chronic back pain [13]. Regenerative cell therapy, which refers to the therapeutic application of stem cells to repair diseased or injured tissue, has received increasing attention from basic scientists, clinicians, and the public. Stem cells hold significant promise for tissue regeneration due to their innate ability to provide a renewable supply of cells that can form multiple cell types, whole tissue structures, and even organs. Stem cells are present in the human body at all stages of life from the earliest times of an embryo through adulthood and senescence. Recently, it could be shown that interactions of human bone marrow-derived stem cells (MSCs) can limit and mitigate the inflammatory responses in the peri-articular fat by promoting anti-inflammatory pathways [14]. Bone marrow-derived MSCs are harvested by puncture of the iliac crest. This procedure is likely accompanied by some risks to the patient, such as long-term damage to the regenerative hematopoietic potential of the bone marrow, after the removal of hundreds of millions of valuable bone marrow cells. About 99% of the cells removed are hematopoietic progenitors, less suitable for tissue renewal outside the hematopoietic system. It seems inefficient to recover a small number of actual pluripotent stem cells to damage the pool of valuable bone marrow cells primarily as it is known that the number of vital bone marrow cells declines with increasing age. A key function of stem cells in the adult body is to contribute to the homeostasis of tissue resident parenchymal cells. As we age, there is a continuous turnover in almost every tissue between dying and replacing cells with the exception of some nerve cells in the brain. For a long time, our body can maintain tissue homeostasis. However, tissue homeostasis can be disturbed with increasing age in all tissues, such as tendons, bone, joints, heart, liver, kidneys, and muscles, in a way that the parenchymal cells, which are responsible for the organ function, are more frequently replaced by mesenchymal fibroblastic cells. This is due to a lack of renewing power, especially if ischemia, infections, accidents, and other inflammatory or traumatic events accelerate the tissue turnover. A good example is chronic wounds that show a number of problems, including insufficient levels of cell proliferation, increased cell senescence/apoptosis, impaired angiogenesis/neovascularization, inflammation, increased production of matrix metalloproteinases (MMPs), increased matrix degradation, and decreased production of extracellular matrix. Stem cell therapy is to be considered as the principal of transferring concentrated stem cells, which have been taken from one part of the body where they are not ‘missed’, to tissue in need of regeneration, in order to re-establish tissue homeostasis. The isolation of stem cells from suitable tissue (such as adipose tissue) and their application to other injured tissue and organs can be interpreted as the most gentle and natural approach to help the body in self-repair by increasing the number of stem cells at a location where they are exhausted but most needed. From these considerations, it also becomes clear that stem cell therapy is not only directed to a specific organ, tissue, or disease, but it will take the function of replacing and repairing tissue and organs that suffer from a lack of repair, renewal, and rejuvenation [15,16].

With the InGeneronTM process, many stem cells can be recovered from adipose tissue [17]. These stem cells from adipose tissue provide three scientifically proven functions: anti-inflammatory effects, antiapoptosis, tissue repair, renewal, and re-placement [18]. These ADRCs can transform into any somatic cell by appropriately exchanging genetic information within the so-called “micro-environment” of the tissue or organ in which they are placed [19]. Consequently, local tissue damage can be treated by injecting stem cells at the site needing repair. Clinical applications have shown that tissue of all three germ lines can be treated with stem cells [15,20,21]. In degenerative diseases of joints, damage to functional tissue induces an inflammatory process, resulting in pain. It has been shown that painful chronic inflammatory disorders of the musculoskeletal system can successfully be treated by stem cell injection [16]. Stem cells, in general, often called MSCs in an inflammatory environment, can influence the immune response by altering cytokine secretion from macrophages, dendritic cells, and T-cell subsets, resulting in a shift from a pro-inflammatory to an anti-inflammatory environment. Successful treatment of degenerative inflammatory processes of the musculoskeletal system, such as facet joints, knee, or shoulder, with injections of adipose tissue-derived stem cells, are well reported in human medicine [22,23]. Adipose Tissue-Derived Regenerative Cells obtained with the InGeneron process contain many so-called vascular-associated pluripotent stem cells [15,19]. These cells can be recovered within about 20 min from a small amount of fatty tissue, without significant harm to the patient.

## 2. Materials and Methods

### 2.1. Inclusion and Exclusion Criteria

All patients were examined and examined and injected by the first author RR. Patients aged 18–80 years with suspected facet joint syndrome and ongoing symptoms of more than one-year duration—despite standard therapeutic measures—were screened for inclusion in the study. A fluoroscopically controlled infiltration of the tissue next to the facet joints by 1 mL (5 mg/mL) ropivacaine was performed in all patients. This was performed to differentiate the source of pain from other causes. Temporary relief of the patient’s pain, an analysis of the patient’s MRI findings, and a thorough clinical examination were used to establish the diagnosis of “Facet joint Syndrome” before the patient could be included in the study. All patients underwent single-level injections on both sides. The test injections were performed using standard fluoroscopy using non-ionic contrast (lohexol 15 g/50 mL). The nonionic contrast medium was characterized by rapid metabolism, so there was little effect of the contrast medium injection on the ropivacaine injection. The patient was positioned in prone position during the procedure.

Exclusion criteria were: other causes of chronic back pain such as acute disc-related pain, spinal stenosis, osteonecrosis, evidence of a malignant neoplasm in the last 24 months, a history of basal cell carcinoma, chronic skin diseases, connective, metabolic, or skin diseases, evidence of an active infection, immunosuppressive medication, renal insufficiency (creatinine > 1, 8 mg/dL) or liver failure (GOT, GPT > 2× typical values, bilirubin > 2 mg/dL). Thirty-seven patients met the inclusion criteria (age 31–78 years, mean age 62.5 years) and were enrolled in the study. Anticoagulation was not an exclusion criterion. If present, anticoagulation was not ceased prior to the liposuction or the test injection.

### 2.2. Tissue Preparation and Treatment

A mini-liposuction was performed under local anesthesia with additional conscious sedation affected by propofol iv injection under the supervision of an experienced anesthesiologist. Prior to the liposuction, the abdominal fat tissue was prepared for the procedure by a subcutaneous injection of a tumescent lidocaine solution. The solution contained 1 g lidocaine and 1 mg epinephrine in 100 mL plus 10 mEq sodium bicarbonate in 10 mL added to 1000 mL of 0.9% physiologic saline for a final lidocaine concentration of 1 g per bag containing 1110 mL or 0.9 g/L (0.09%) 500 mL were injected bilaterally in the umbilical region. The solution remained 10 min in the subcutaneous tissue before we proceeded with the liposuction. The liposuction was performed using a normal 50 cc syringe and a standard liposuction needle. The tissue harvest was performed by hand. No mechanical suction device was used. An amount of 50–100 mL of abdominal fat tissue was removed by this mini-liposuction and processed using a commercially available system. To recover stem cells from adipose tissue, we use the CE-certified Transpose RT™ system from InGeneron GmbH. The removed adipose tissue sample was transferred into InGeneron’s processing tube, and Matrase™ enzymatic reagent was added to the sample before it was processed in the Transpose^®^ Tissue Processing Unit. During the first run, the extracellular enzymatic degradation by the specific Matrase TM enzyme liberates the regenerative cells from their surrounding matrix. Three subsequent filtering and washing steps are performed to separate and purify the regenerative cells from fat and adipocytes, remaining enzymes, and cell debris. Within 60 min, purified regenerative stem cells recovered from the patient’s abdominal fat tissue were ready for immediate use in the same patient as an autologous transplant Figure 1.

### 2.3. Cell Yield, Cell Viability, Number of Living Cells per mL Lipoaspirate and Cell Size

Due to the results of a previous study, we can describe the cell yield, cell viability, number of living cells per ml lipoaspirate, and cell size detail [24]. Compared to Transpose RT/no Matrase isolation, Transpose RT/Matrase isolation of ADRCs from lipoaspirate resulted in the following, statistically significant differences in the final cell suspension (all values given as mean ± SEM): approximately nine times higher cell yield (7.2 × 105 ± 0.90 × 105 Transpose RT/Matrase-isolated ADRCs per ml lipoaspirate vs. 0.84 × 105 ± 0.10 × 105 Transpose RT/no Matrase-isolated ADRCs per ml lipoaspirate; *p* < 0.001; n = 12 matched pairs of samples); approximately 41% higher mean cell viability (85.9% ± 1.1% in case of Transpose RT/Matrase-isolated ADRCs vs. 61.7% ± 2.6% in case of Transpose RT/no Matrase-isolated ADRCs; *p* < 0.001; n = 12 matched pairs of samples); approximately twelve times higher mean number of living cells per ml lipoaspirate (6.25 × 105 ± 0.79 × 105 Transpose RT/Matrase-isolated ADRCs per ml lipoaspirate vs. 0.52 × 105 ± 0.08 × 105 Transpose RT/no Matrase-isolated ADRCs per ml lipoaspirate; *p* < 0.001; n = 12 matched pairs of samples each). Of importance, the mean relative number of viable cells obtained by Transpose RT/Matrase isolation (85.9%) exceeded the proposed minimum threshold for the viability of cells in the SVF of 70% established by the International Federation for Adipose Therapeutics and Science (IFATS), whereas the mean relative number of viable cells obtained by Transpose RT/no Matrase isolation (61.7%) did not. The difference in mean cell diameter between Transpose RT/no Matrase-isolated ADRCs (10.2 μm ± 0.1 μm) and Transpose RT/Matrase-isolated ADRCs (10.6 μm ± 0.1 μm) was only approximately 4% and did not reach statistical significance (*p* = 0.05; n = 12 matched pairs of samples). Accordingly, both the number and viability of cells in the final cell suspension were statistically significantly higher after Transpose RT/Matrase isolation of ADRCs from human adipose tissue than after Transpose RT/no Matrase isolation. ASCs derived from Transpose RT/Matrase-isolated ADRCs formed on average 16 times more CFUs per ml lipoaspirate (4973 ± 836; mean ± SEM) than ASCs derived from Transpose RT/no Matrase-isolated ADRCs (307 ± 68) (*p* = 0.002; n = 10 matched pairs of samples).

### 2.4. Injection of the ADRCs

The cellular preparation of these fresh, autologous ADRCs was injected under para-articular fluoroscopic control on both sites of the individually affected facet joints. The joints in the adjacent segments were injected as well as the target segment (typically, L3 to L5 and L5/S1). The procedure was performed within the same session of fat tissue recovery, and preparation of the cells. Stem cell suspension was diluted with physiological saline to 1 mL per injected joint. At the Adipose-derived stem cell injection, contrast media was not employed due to unknown interaction with the regenerative cells. The injection was performed in prone position in the same fashion as the test injection. However, mild sedation was used for the final injection. The fat removal to injection procedure was typically finished in two to three hours (Figure 2). Patients were discharged from the outpatient clinic within four hours after beginning of the treatment. Pain medication was not recommended and not employed by the study population.

### 2.5. Follow-Up

According to the study protocol, the primary endpoints of the present study were long-term safety, as indicated through the rate of treatment emergent adverse events (TEAEs), efficacy of pain and function. All 37 patients completed the follow up examination and questionnaire. None of the study participants was lost during follow-up. As part of the follow-up, the Visual Analogue Scale (VAS) for back and leg pain and the Oswestry Disability Index (ODI) were recorded before treatment and postoperatively at one week, one year, and five years after the treatment. Kind, type, severity, and duration of any adverse events were recorded at each postoperative visit.

## 3. Results

### 3.1. Preoperative Symptoms

The most common preoperative symptom was local back pain (in 100% of cases). Before treatment, none of the patients had a motor deficit. One patient had suffered from a sensory deficit at L5 on the right side (2.7%) after six previous operations on the lumbar spine. No other neurological symptoms or signs were present during the initial examination. Clinical evaluation showed a compression pain of the facet joints in the affected area during the physical examination in all patients. Only lumbar pain syndromes were included in the evaluation. No one of the study participants suffered from cervical facet joint syndrome. All patients underwent MRI examinations. In the preoperative MRI, nerve root compression, acute fractures, or spondylodiscitis was ruled out in all patients. All patients showed degenerative changes in facet joints of varying degree. In seven patients (18.9%), there were radiological signs of osteochondrosis Modic I and II (Figure 3). Spondylolisthesis was not present in any of the study participants.

### 3.2. Pain Levels and Oswestry Disability Index

The preoperative pain level measured by the Visual Analog Scale means level was VAS 6.8 with an individual range of 5–10 (on a scale of 0 = no pain, 10 = maximum pain), and the Oswestry Disability Index (ODI) was assessed at 71.05% (range 43–91). After one week, patients already reported an improvement reflected by a decreased Visual Analog Scale pain level. At the one-year follow-up (mean 13.2 months, range 12–16 months), the mean VAS level was 1.5, and the mean ODI was 17.5%. At the five-year follow-up (mean 61.7 months), the mean VAS level was 1.4, and the mean ODI was 18.7%. Every patient reported improved VAS pain after treatment with ADRCs compared to the baseline (Figure 4 and Figure 5). This result was observed in the short term (one week and one month) and in the longer term after five years of follow-up.

### 3.3. Complications

After liposuction, one patient developed a relatively sizeable subcutaneous hematoma of about 10 × 15 cm at the liposuction site. The patient was on anticoagulation for cardiac arrhythmia. The subcutaneous hematoma was treated conservatively. It caused local pain for 17 days and resorbed spontaneously without further consequences. There were no other minor or severe complications, in particular, no infections and no local or systemic minor or significant adverse events related to the procedure or the sedation.

## 4. Discussion

Low back pain is a very common disabling condition and has an enormous economic impact on ageing societies. Around 70% of all people will suffer at least once in their lifetime from severe low back pain, as about 25% of all people suffer from chronic lumbago. Causal therapy is unknown, and regular use of NSAIDs often leads to unfavorable comorbidities and is associated with considerable socio-economic costs. Corticosteroid injections are useful in the initial phase of the disease but are known to lose the beneficial effect overt time. Ablation procedures show a temporary effect. In the current study, we report initial clinical results of ADRC treatment and five-year follow-up in patients with chronic facet joint syndrome.

In the context of degenerative lumbar spine diseases, pain is primarily due to inflammatory reactions resulting from degenerative changes. Inflammatory cytokines have been demonstrated in the facet joint [4] to produce arthropathic changes in degenerative lumbar spine disease. As a result, pain is linked to chemical factors associated with inflammatory responses and cytokines. This theory corroborates the better clinical effects of nonsteroidal anti-inflammatory drugs and corticosteroids over opioids or other analgesics [9]. The immune-modulatory impact of stem cells is, therefore, of great interest for the prevention of inflammatory reactions in degenerative diseases of the musculoskeletal system. ADSCs exhibit potent immunosuppressive and anti-inflammatory activities, and exosomes were shown to play an important role in these processes. In recent years, apoptotic bodies, a major class of extracellular vesicles released as a product of apoptotic cell disassembly, have become recognized as another key player in immune modulation. Successful treatment has also been reported with experimental autoimmune disorders such as collagen-induced arthritis and multiple sclerosis. However, conflicting results on how immune modulation is achieved by stem cells are still discussed [7].

Our study used adipose tissue stem cells (ADSCs) because several prior studies have found a significant qualitative difference between bone marrow stem cells and adipose tissue stem cells. The main difference is the significantly higher number of vascular-associated pluripotent stem cells found in adipose tissue, plus the more facilitated access to adipose tissue compared to bone marrow [15,17]. Harvesting a small amount of fatty tissue damages the patient less than removing millions of valuable bone marrow cells. Bone marrow yields a relatively low number of genuinely regenerative cells. Culturing these cells (MSCs) before use in clinical applications is customary, leading to gene expression changes [25]. Adipose tissue-derived stem and regenerative cells can be isolated and administered in large numbers at the “point of care” without cell expansion/culturing [26]. These non-engineered cells also lead to high clinical safety and efficacy [20,21,27].

Our data suggest an initial anti-inflammatory impact of the cells used. We also clinically observed an acute anti-inflammatory effect of the ADSCs, evidenced by the first clinical improvement often seen already 48 h after treatment. This result suggests an additional therapeutic benefit for other degenerative diseases of the musculoskeletal system. Like regulatory T cells (TREG), ADSCs can migrate to joints where they can act locally within the inflamed synovium to reduce immune cell proliferation and function through the secretion of soluble inhibitory factors. They can also systematically suppress the host immune response through a shift in the T1/T2 cell balance, suggesting that ADSC-induced immune modulation is not only mediated by a single mechanism, indicating critical therapeutic applications well beyond the field of degenerative joint disease. Further studies should help better understand stem cells’ underlying mechanisms of immune modulation and how stem cells derived from adipose tissue exert the same function in a new location as they do in the initial site of adipose tissue from which they were harvested [24,28].

## 5. Conclusions

The current study is, to the best of our knowledge, the first publication in the scientific literature of a 5-year follow-up long-term assessment evaluating the treatment of chronic facet joint syndrome with autologous adipose tissue-derived stem cells. All patients in this study served as their internal paired control. The authors’ results warrant further investigation of ADRCs for facet joint syndrome treatment, especially because this condition has a high economic impact, no causal therapy is available, and it is a disabling condition in a significant portion of the population. Future randomized prospective controlled trials assessing the safety and efficacy of ADRC treatment in comparison to the best-known standard of care interventions should be conducted to corroborate the results of this long-term observational study.

## Figures and Tables

**Figure 1 jpm-13-00436-f001:**
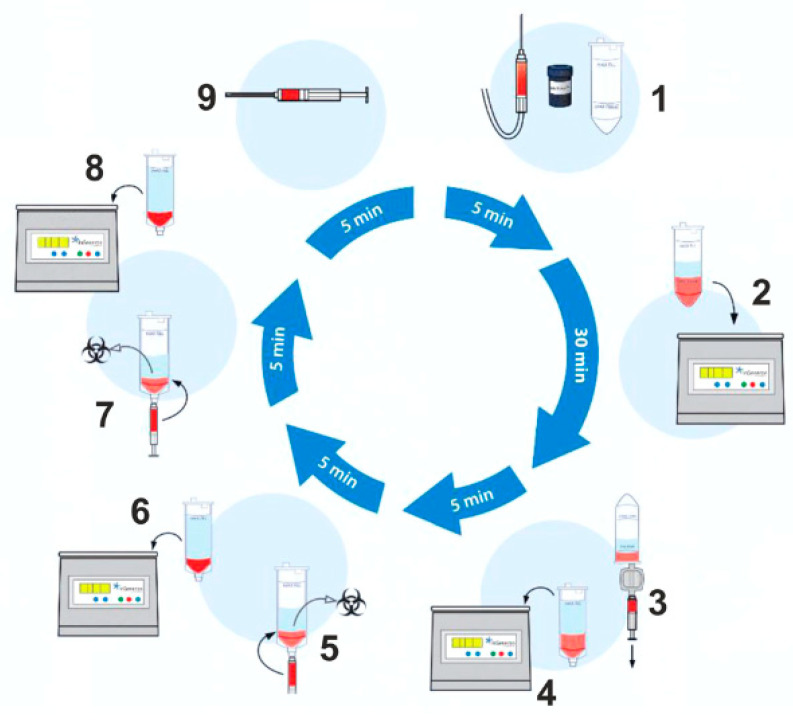
Schematic overview of the lipoaspirate processing procedure. The tissue sample is transferred into the processing tube of the Transpose^®^ Ultra Lipoaspirate Processing System, and the proprietary Matrase™ enzymatic reagent is added to the sample before it is inserted into the Transpose^®^ Ultra Tissue Processing Unit. During the first run, efficient enzymatic extracellular degradation liberates the regenerative cells from their surrounding matrix. Subsequent washing and filtering steps isolate these cells before their immediate therapeutic application. The process shown below takes approximately 60 min after lipoaspiration.

**Figure 2 jpm-13-00436-f002:**
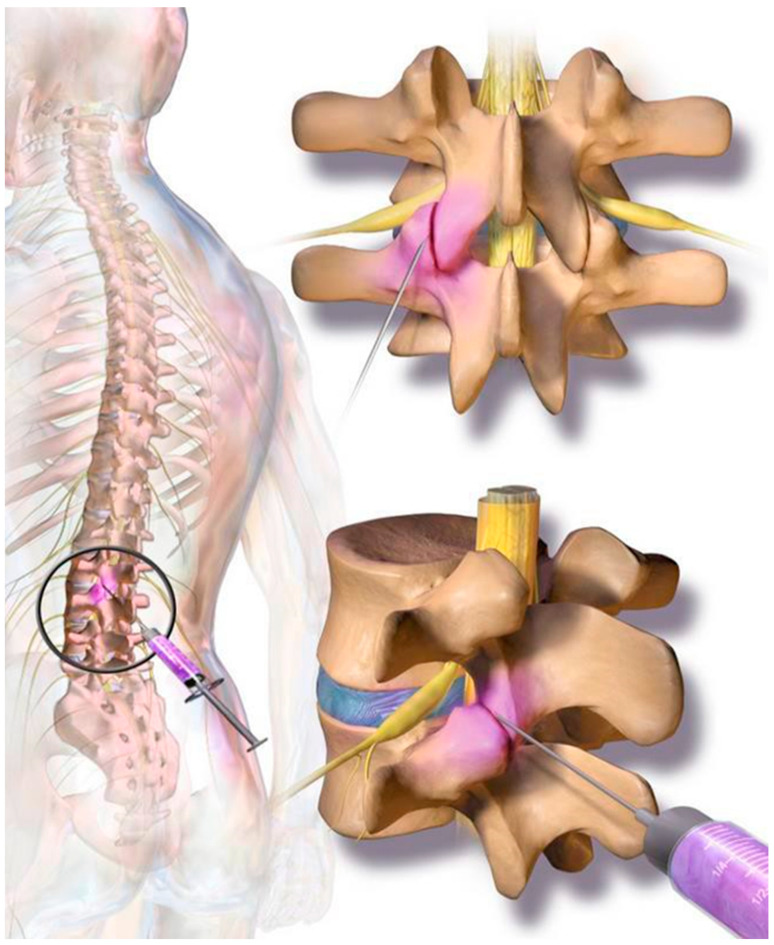
On each level at the back of the spine, two small facet (zygapophyseal) joints—one from an upper and one from a lower vertebra—act as connectors to the spine and provide support as they allow the spine to bend and move. Facet joint syndrome significantly contributes to the high prevalence of back pain observed in western societies. Current therapies include local cortisone injections; however, there are mixed and negative reports regarding long-term efficacy for facet joint syndrome pain relief.

**Figure 3 jpm-13-00436-f003:**
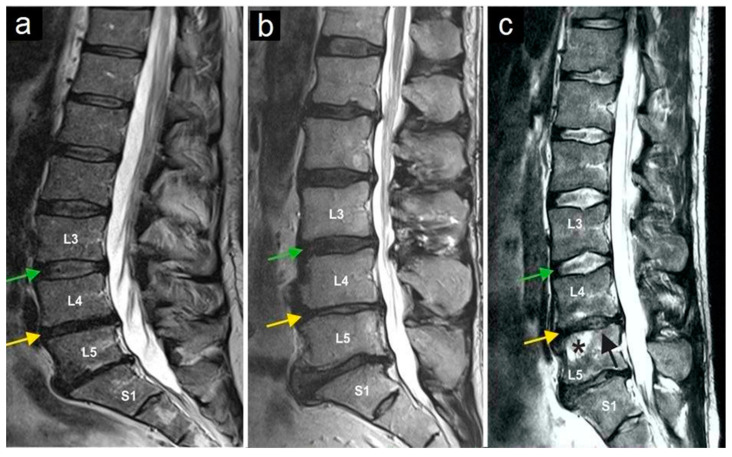
Magnetic resonance imaging of three lumbar spine patients (**a**–**c**). The green arrows mark the L3/L4 disc, and the yellow arrows mark the L4/L5 disc. Note the normal structure of the L3/L4 disc, while the yellow arrows indicate a reduced intervertebral disc height (and thus the disc space) at L4/L5, as observed in all three patients. The star (**c**) marks bone marrow edema in vertebral body L5, and the black arrow (**c**) indicates the collapse of the superior endplate of the L5 vertebral body.

**Figure 4 jpm-13-00436-f004:**
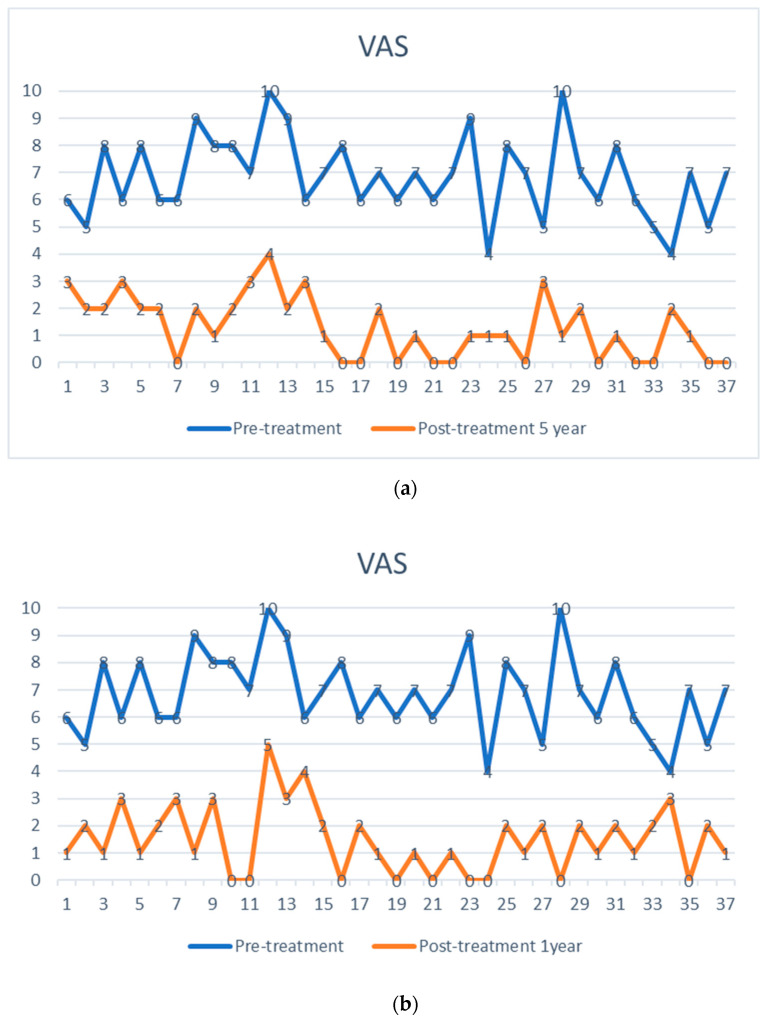
Visual analog pain score (VAS) of 37 facet joint pain patients, shown as pre-treatment and post-treatment VAS score with adipose-derived regenerative cells at one week (**a**), one year (**b**), and five years (**c**). The comparison is shown before treatment (upper points in blue), (**a**) after one week, (**b**) after one year, and (**c**) after five years (lower points in orange). Each patient serves as their internal control.

**Figure 5 jpm-13-00436-f005:**
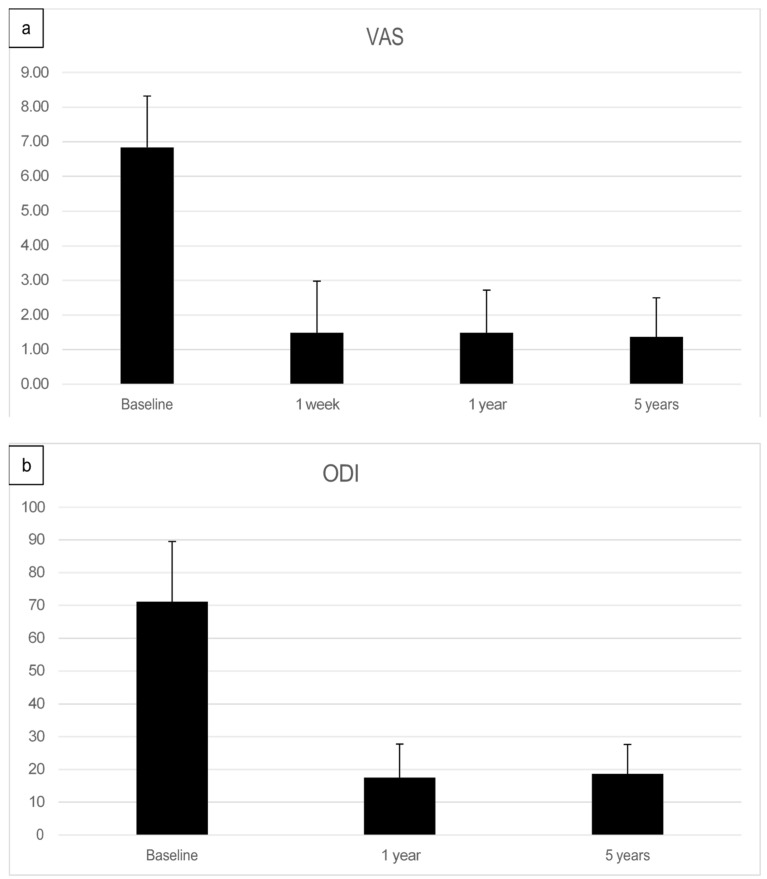
(**a**) Summary of Visual analog pain score (VAS)depicted as a bar graph of facet joint patients treated with adipose-derived regenerative cells at baseline, after one week, after one year, and five years post-treatment. (**b**) Mean Oswestry Disability Index (ODI) score pre-treatment, at one year, and five years post-treatment.

## Data Availability

Data is available upon request from the corresponding author.

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
