# Peer review of "Safety and Efficacy of Autologous Stem Cell Treatment for Facetogenic Chronic Back Pain"

_jpm, 2023, doi:10.3390/jpm13030436_

Round 1
Reviewer 1 Report
This study is very interesting and the use of Unmodified Adipose Tissue Derived Regenerative Cells (ADRC) it is a promising application for facetogenic chronic back pain. In the light of evidence that the adipose tissue was enzymatically manipulated by The Matrase Reagent (enzyme blend of collagenase and neutral protease) AA should clarify if this treatment is considered advanced therapy medicinal products (ATMPs) for The European Medicine Agency and the U.S. Food and Drug Admin- istration (FDA)
Author Response
Dear reviewer pleas find enclosed our revised manuscript.
Best regards
Ralf Rothörl
Reviewer 2 Report
Minor
1 Please list the basic information of included patient with the clinical efficacy
2 Please defined “adipose tissue derived regenerative cells”. Mesenchymal stem cells? Vascular cells? Neuronal cells? Or mixture of them?
Author Response

(The authors gave the same response as above.)
